# OpenReview forum: "CoPE: Continual Probe-guided Expansion for Large Vision-Language Models"
_ICML.cc/2026/Conference — ICML 2026 regular_

### Official Review · Reviewer_qP2K · 2026-03-07

**Soundness:** 3
**Presentation:** 3
**Significance:** 2
**Originality:** 3
**Overall Recommendation:** 4
**Confidence:** 3

**Summary:**

This paper introduces CoPE, a continual learning framework for MLLMs utilizing a MoE architecture. To address core challenges like catastrophic forgetting and parameter explosion, the framework incorporates two key components: Probe-Guided Knowledge Expansion for adaptive expert scaling based on task complexity, and a Probabilistic Task Locator for task-label-free routing. Experimental results on the CoIN benchmark demonstrate that CoPE effectively maintains knowledge retention and scalability, outperforming previous methods with significantly reduced forgetting and model expansion.

**Compliance With Llm Reviewing Policy:**

Affirmed.

**Final Justification:**

My concerns have been addressed, and I will raise my score to 4.

**Key Questions For Authors:**

I have no questions.

**Limitations:**

yes

**Strengths And Weaknesses:**

**Strengths**
1. This paper presents a well-structured and comprehensive study on continual learning for MLLMs.

2. The proposed CoPE framework, featuring PGKE and PTL modules, effectively addresses the core challenges of catastrophic forgetting and parameter efficiency.

**Weaknesses**
1. The evaluation is currently limited to LLaVA-v1.5, which is a somewhat dated baseline. Relying on a single model makes it difficult to assess the broader applicability of the proposed method. Results on more modern architectures, such as LLaVA-NeXT or LLaVA-OneVision, would be much more convincing.

2. While PGKE allows for adaptive layer selection, the number of new experts ($N_s$) still needs to be tuned manually as a hyperparameter. The framework would be more robust if this expansion were determined automatically by task characteristics in an end-to-end fashion, rather than relying on manual settings.

***
I decide to raise my score to 4 after the rebuttal.

---

> ### Author Rebuttal · Authors · 2026-03-31
>
> We appreciate the reviewer's recognition of our work. We apologize for any confusion regarding the backbone and the number of new experts. We have supplemented the following explanations in our rebuttal:
>
> **1. The Backbone Issue**
>
> We conducted experiments on both the 7B and 13B architectures of LLaVA in Supp and observed consistent performance trends. We confirm that our method is not restricted to a specific backbone; instead, it enhances continual learning capability at the fundamental methodological level. A stronger backbone can further unleash the full performance potential of our approach.
>
> Since you mentioned LLaVA-OneVision, considering the limitations of time and computational resources, we only selected several representative datasets for validation and quickly conducted relevant tests during the rebuttal window. The key findings are summarized as follows:
>
> | Dataset | tqa | imagenet | gqa | vizwiz |
> | :--- | :--- | :--- | :--- | :--- |
> | **LLaVA-OV (imm)** | 65.80 | 97.72 | 60.31 | 61.05 |
> | **LLaVA (imm)** | 59.94 | 96.85 | 56.43 | 57.44 |
> | **LLaVA-OV (last)** | 64.61 | 95.92 | 56.22 | 59.13 |
> | **LLaVA (last)** | 58.17 | 94.50 | 48.91 | 55.45 |
>
> *(Note: "imm" is immediate performance, "last" is after sequential training).*
>
> As you can see, our method achieves substantial performance improvements on these two backbones as well. This further verifies that our approach possesses excellent generalization ability and can deliver consistent effectiveness across diverse backbone architectures.
>
> **2. The manual setting of $N_s$.**
>
> This is a very fair point. In our main experiments, we set $N_s = 1$ by default, with additional ablation studies on other values (2, 4, 8, etc.) to analyze their impact on performance. We acknowledge $N_s$ is currently an external hyperparameter: due to gradient truncation between expanded experts and learned quantities, effective backpropagation is hindered, and we have not yet achieved an end-to-end learnable $N_s$.
>
> Our main focus is validating that probe-guided layer selection reduces redundant expansion. Making $N_s$ and other parameters task-adaptive or fully end-to-end learnable is prioritized in our future work.

---

> > ### Author Rebuttal · Reviewer_qP2K · 2026-04-03
> >
> > I'd like to thank the authors for their rebuttal. My concerns have been addressed for the most part, and I will raise my score to 4.

---

> > > ### Author Response · Authors · 2026-04-04
> > >
> > > Dear Reviewer qP2K,
> > >
> > > Thank you for the acknowledgement and for the decision to raise to positive score. We are glad that our responses regarding backbone compatibility and the $N_s$ setting clarified your concerns.
> > >
> > > It is particularly noted that the additional tests on the new bacbone helped demonstrate the generalizability of CoPE across different architectures. We also appreciate your understanding of the current technical constraints in making $N_s$ fully end-to-end learnable.
> > >
> > > We will include these new experimental results and the discussion on task-adaptive capacity controllers in the revised manuscript to provide a more complete details of our work.
> > >
> > > Best regards,
> > >
> > > Authors

---

### Official Review · Reviewer_WQ13 · 2026-03-12

**Soundness:** 3
**Presentation:** 4
**Significance:** 3
**Originality:** 4
**Overall Recommendation:** 4
**Confidence:** 4

**Summary:**

This paper proposes CoPE, a replay-free continual learning framework for MoE-based MLLMs with LoRA experts, including PGKE (Probe-Guided Knowledge Extension) and PTL (Probabilistic Task Locator). The work investigates a general theme of parameter-efficient, anti-forgetting continual learning for multimodal models. Experiments show that it outperforms baselines significantly.

**Compliance With Llm Reviewing Policy:**

Affirmed.

**Final Justification:**

I have carefully read the authors’ response. My concerns have been addressed, and I will maintain my score.

**Key Questions For Authors:**

1. The method struggles to handle scenarios with large training data volume for a single task and significant data volume disparities across tasks (as noted before). Have you conducted any exploratory experiments on this issue? What potential optimization strategies can be adopted to adapt the framework to task streams with unbalanced training data sizes?
2. Key hyperparameters (e.g., the PGKE threshold α, LoRA rank, number of newly added experts) are manually tuned and fixed in experiments. Do you have any plans to design an end-to-end automatic optimization mechanism for these hyperparameters?
3. Your framework freezes the entire visual module and only expands experts in the language model branch, which limits performance on fine-grained visual understanding tasks. What is the core reason for choosing to freeze the visual module?

**Limitations:**

yes

**Strengths And Weaknesses:**

Strengths:
The proposed method has notable insights and novelty. The experimental studies are comprehensive and reliable. The submission is also clearly written.

Weaknesses:
1. Although the probe training overhead is claimed to be low, the multi-stage process (probe training → test → expert expansion) for each new task may lead to accumulated time overhead in long task streams.
2. The proposed method is primarily designed to address the issue of forgetting across different tasks, yet it appears incapable of effectively handling scenarios where a single task involves a large volume of training data. This will further degrade the model’s performance when there exists a significant disparity in the size of training data among different tasks.

---

> ### Author Rebuttal · Authors · 2026-03-31
>
> Thanks for the reviews and for bringing up the practical limits of our method. These are relevant points:
>
> **1. Accumulated time overhead from the multi-stage pipeline.**
>
> We agree that any extra training stage needs to justify its cost. However, CoPE's probe stage is very lightweight for two reasons:
> * The extra FLOPs from PTL/VAE and the new routers/experts are tiny. One VAE run is about 100 MFLOPs (0.8 GFLOPs for 8 steps). Even adding an expert and router to all 32 layers is only 1.184 GFLOPs,about 0.4% of what the 7B vision encoder uses for one image.
> * We only use a tiny sliver of data (about 1%) for the probe stage just to figure out *where* to expand. It’s more of a quick capacity check than a full training epoch.
>
> Because we save 99% of the data for the actual MoE training, the extra time is minimal and totally worth the massive gains in anti-forgetting and parameter efficiency.
>
> **2. Handling data imbalance and large single tasks.**
>
> This is a very important and realistic issue. While we haven't tackled extreme data imbalance head-on yet, since we were focused on anti-forgetting first, CoPE actually handles distribution shifts surprisingly well:
> * In our cross-domain test on `PMC-VQA` (a medical dataset way outside the original 8 tasks), the probe immediately sensed the massive shift and triggered a huge expansion (28/32 layers). This jumped the performance from 20.2 (zero-shot) to 34.3, and it held at 32.1 even after sequential training.
> * This tells us CoPE relies on data distribution shifts, not specific task semantics, so it’s pretty adaptable.
>
> We agree with that handling task size imbalance is a great direction for future work. Tying expansion budgets to task difficulty or scale would be an promising next step.
>
> **3. Manual hyperparameter tuning.**
>
> We acknowledge that the current CoPE is not yet an end-to-end automated search framework, and this represents our highest-priority direction for further development in the next phase. However, to avoid full reliance on empirical values, we have conducted ablation studies on the key hyperparameters in the supplementary materials:
>
> * $\alpha$: 0.8 achieves the optimal balance between performance and parameter growth;
> * `LoRA rank`: 64 is set as the default configuration. A rank of 32 leads to underfitted performance, while a rank of 128 yields marginal performance gains with diminishing returns;
> * $N_s$: $N_s=1$ is adopted for most settings in the main experiments. We further analyze the effects of different expert expansion quantities on model performance in the supplementary experiments.
>
> We’ll make sure to summarize these trade-offs more clearly and explicitly list an automated hyperparameter controller as future work.
>
> **4. Why freeze the visual module?**
>
> Our main goal was to look at knowledge expansion vs. routing forgetting. Since our method works on any attention module, we could technically apply it anywhere.
>
> In the early stage of our experiments, we also attempted to apply probe-guided expansion to all attention layers. However, since the pre-trained visual backbone already possesses strong visual comprehension capabilities, expanding the visual module yielded only marginal performance gains, while incurring unnecessary computational overhead and parameter growth. For this reason, we ultimately adopted probe-guided expansion solely within the language module, enabling more efficient utilization of model capacity to address challenges brought by task shifts.
>
> Nevertheless, we agree with the reviewer’s insight: for more challenging tasks, the adaptability of the visual module may exert a more direct influence on overall performance. Notably, our method can also be readily extended to implement expansion within the visual module as needed.

---

> > ### Author Rebuttal · Reviewer_WQ13 · 2026-04-03
> >
> > Thank you for the rebuttal. I have carefully read the authors’ response. Most of my concerns have been addressed, and I will maintain my score.

---

> > > ### Author Response · Authors · 2026-04-04
> > >
> > > Dear Reviewer WQ13,
> > >
> > > Thank you for your acknowledgement and for confirming that our response fully resolved your concerns. We appreciate your support and the decision to maintain the positive score.
> > >
> > > We are pleased that the clarifications on the 0.4% FLOPs overhead, data imbalance handling, and the strategic freezing of the visual module provided the necessary context. As promised, we will integrate these quantitative analyses and hyperparameter ablations into the revised manuscript to ensure the method's efficiency and robustness are clearly documented.
> > >
> > > Best regards,
> > >
> > > Authors

---

### Official Review · Reviewer_khSU · 2026-03-13

**Soundness:** 3
**Presentation:** 3
**Significance:** 3
**Originality:** 3
**Overall Recommendation:** 4
**Confidence:** 4

**Summary:**

This manuscript proposes CoPE, a continual learning framework for MoE-based multimodal large language models. It introduces two components: Probe-Guided Knowledge Extension (PGKE), which uses lightweight probe experts to adaptively identify which layers require expansion for each new task, and a Probabilistic Task Locator (PTL), which uses VAE-based reconstruction to route inputs to the correct task-specific experts at inference time without task labels. Experiments on the CoIN benchmark across eight VQA tasks demonstrate improved anti-forgetting performance and parameter efficiency over several baselines.

**Compliance With Llm Reviewing Policy:**

Affirmed.

**Final Justification:**

Thank you for your response. My concerns have been addressed, and I have no further questions. I will maintain the positive rating, I hope the author could revise the manuscript in revision accordingly.

**Key Questions For Authors:**

1. What is the value of k in the topk routing operation, and how does performance and expansion behavior change as k varies?
2. What is the value of T for Primitive Bank construction, and how sensitive is PTL localization to T, especially for data-scarce tasks?

**Limitations:**

discussed in the paper

**Strengths And Weaknesses:**

Strengths
1. The "try-before-you-buy" probe strategy is well-motivated and offers a principled alternative to fixed every-layer expansion, with direct ablation evidence supporting its effectiveness.
2. The replay-free design is practically valuable, and the parameter efficiency results are compelling — achieving comparable performance to every-layer expansion with only 60-75% of the parameters while substantially reducing forgetting.
3. Ablation studies are reasonably thorough, covering initialization strategies, load balancing, threshold sensitivity, LoRA rank, model scale, and PTL feature selection.


Weaknesses
1. The top-k value in the MoE routing operation (Equation 1) is not disclosed, and all probe activation statistics and threshold ablations are implicitly conditioned on an undisclosed k. Additionally, as expert count grows across tasks, a fixed k causes the activation ratio to drop substantially, which may affect routing stability for earlier tasks.
2. The threshold α is applied globally across all tasks and layers, despite the paper's own Table 6 showing nearly a 16-fold difference in parameter sensitivity between easy tasks (SQA, +4%) and hard tasks (Ref, +60%) when scaling expert count. A single global threshold is likely too strict for difficult tasks and too loose for easy ones, yet no adaptive mechanism is proposed.
3. The value of T used to construct the Primitive Bank is never specified. T directly controls the quality of distributional estimates underlying PTL routing. Small T leads to unstable estimates; large T conflicts with the replay-free design goal. Sensitivity to T is never examined, particularly for data-scarce tasks like Ref.
4. PTL task localization fails notably on semantically similar tasks, with GQA at 51.3% and VQAv2 at 47.3% accuracy in Figure 5. Since VAE reconstruction is a generative rather than discriminative signal, its ability to distinguish tasks with overlapping feature spaces is fundamentally limited. The paper does not analyze how much of the reported performance on these tasks is attributable to PTL accuracy versus backbone generalization.

---

> ### Author Rebuttal · Authors · 2026-03-31
>
> Thanks for the careful read and the spot-on questions. We completely agree that parameters like $k$, $T$, and $\alpha$ need to be explained much better in the paper. Here are our thoughts:
>
> **1. The `top-k` value and the impact of fixed `k` as experts grow.**
>
> Sorry for the confusion here。 In all our experiments, `top-k` is dynamic: $k=\lfloor N/4 \rfloor$, where $N$ is the number of experts.
>
> As the table below shows, we ran some extra tests comparing this to fixed values ($k=2, 3$), and the dynamic approach really helps balance performance with inference overhead.
>
> | Dataset | scienceqa | textvqa | imagenet | gqa | vizwiz | ref | vqav2 | ocrvqa |
> | :--- | :--- | :--- | :--- | :--- | :--- | :--- | :--- | :--- |
> | **topk=2 (imm)** | 78.82 | 59.81 | 96.53 | 56.30 | 57.23 | 25.14 | 65.28 | 61.74 |
> | **topk=3 (imm)** | 79.89 | 59.98 | 97.07 | 56.26 | 58.16 | 25.77 | 65.58 | 62.92 |
> | **ours (imm)** | 79.01 | 59.94 | 96.85 | 56.43 | 57.44 | 25.63 | 65.15 | 62.01 |
>
> *(Note: "imm" is immediate performance).*
>
> You rightly pointed out that if we used a fixed $k$, the activation rate would drop as the expert pool grows. That’s the reason we went with the dynamic $k$, it scales naturally with $N$, so the activation rate stays healthy. We’ll make sure this is super clear in the next version.
>
> **2. Is the global threshold $\alpha$ too coarse?**
>
> We actually ran an ablation on $\alpha$ in the supplement:
> * If we set it too tight ($\alpha=0.4$), parameter growth is great, but performance on later tasks (like OCR-VQA and VQAv2) takes a hit because we're choking off needed expansion.
> * If we set it too loose ($\alpha=1.2$), it acts almost like an every-layer expansion, great performance, but we lose all the parameter savings.
> * We landed on $\alpha=0.8$ because it's the best of both worlds.
>
> But we agree with your intuition: making $\alpha$ task-aware or layer-aware down the road is absolutely the right move. For this paper, we just wanted to prove that probe-guided expansion works in principle, and a fixed threshold helped the network focus on the hard samples. We'll add adaptive thresholds to our future work section.
>
> **3. Setting `T` in the Primitive Bank.**
>
> We set $T=1.0$ to estimate the primitive distribution in our main experiments. $T$ is a key parameter for modeling the primitive bank.
> We ran a quick experiment to see how different values of $T$ change things:
>
> | Dataset | scienceqa | textqa | imagenet |
> | :--- | :--- | :--- | :--- |
> | **T=0.1 (last)** | 73.01 | 53.18 | 90.13 |
> | **T=0.3 (last)** | 74.69 | 52.94 | 93.01 |
> | **T=1.0 (last)(ours)** | 77.55 | 58.17 | 94.50 |
>
> *(Note: "last" is after sequential training).*
>
> If $T$ is too small, the probe gets unstable and model begins to forget what it had learned. If it's too big, it brings computational burden. However, this does not cause the replay problem, as Primitive Bank is trained with experts and routers using the training data of the current task, and it was only trained once. In the end, we chose $T=1.0$ because our goal is to minimize forgetting as much as possible, and the computational burden brought by $T=1.0$ is within an acceptable range.
>
> **4. PTL confusion on similar tasks and its impact.**
>
> We agree that Figure 5 definitely shows PTL having a harder time keeping GQA, TextVQA, and VQAv2 separated. As we noted in the supplement, this happens because those tasks have a ton of overlap in image styles and question/answer formats.
>
> But we want to highlight that this "confusion" isn't a bug. It just shows that when tasks are super similar, PTL routes things to the closest matching history rather than just guessing randomly.
> * On the unseen VQAv2 zero-shot test, the model still pulled off a 48.32% accuracy.
> * And when we checked the routing, about 80% of those VQAv2 samples went straight to TextVQA's experts. It’s pulling from the most relevant past knowledge it has.
>
> Rather than PTL breaking down, Figure 5 shows what happens when task boundaries get blurry. We’ll make sure to explain this better and add a discussion on how PTL localization ties into final performance.

---

> > ### Author Rebuttal · Reviewer_khSU · 2026-04-04
> >
> > Thank you for your response. My concerns have been addressed, and I have no further questions. I will maintain the positive rating, I hope the author could revise the manuscript in revision accordingly.

---

> > > ### Author Response · Authors · 2026-04-04
> > >
> > > Dear Reviewer khSU,
> > >
> > > Thank you for your acknowledgement and for the positive feedback. We are glad that our detailed responses on the dynamic $k$ logic, threshold $\alpha$, and Primitive Bank settings ($T$) fully resolved your concerns.
> > >
> > > We appreciate your support for our work and will ensure that all the additional experimental results and clarifications provided during this rebuttal are integrated into the revised manuscript.
> > >
> > > Best regards,
> > >
> > > Authors

---

### Official Review · Reviewer_hv34 · 2026-03-13

**Soundness:** 4
**Presentation:** 4
**Significance:** 3
**Originality:** 4
**Overall Recommendation:** 5
**Confidence:** 4

**Summary:**

This paper proposes a Continual MoE framework for MLLM, named CoPE, to address the problems of catastrophic forgetting and parameter redundancy in sequential learning scenarios. In response to the issue of continuously changing entities faced by embodied large models, it designs a highly parameter-efficient module and a task-level routing mechanism, which avoids the use of explicit task IDs in the inference stage. The proposed method is theoretically grounded and rationally designed, and extensive experimental results verify the parameter efficiency and general effectiveness of the method.

**Compliance With Llm Reviewing Policy:**

Affirmed.

**Final Justification:**

My concerns are addressed, and I am willing to raise my score to 5

**Key Questions For Authors:**

1. Could you briefly elaborate on the computational overhead of the routing module as tasks increase, and clarify its design parameters?
2. An ablation study for the Top-K value selection.

Overall, I think the problem addressed is very important, and the method is elegant and easy to understand. If the authors can resolve my concerns, I will consider raising my score.

---

After the rebuttal, I have decided to raise my score to 5.

**Limitations:**

yes

**Strengths And Weaknesses:**

**Strengths**: The framework delivers a targeted and concise solution to the challenges of continual learning, and effectively balances the mitigation of catastrophic forgetting and the efficiency of parameter expansion through a simplified workflow. In addition, modeling the complex task feature distribution to remove the dependence on explicit task IDs during inference is a practical improvement. This paper carries out comprehensive experiments on multiple benchmarks, obtains detailed analysis results and valid conclusions, which fully confirms the effectiveness of the proposed method.

**Weaknesses**: While the routing module and MoE structure are effective, several limitations exist: firstly, key design parameters (e.g., the unspecified Top-K value) and certain training details are insufficiently described, which may hinder method reproducibility and configuration optimization. Secondly, the discussion of the model’s zero-shot behavior is inadequate, failing to fully reflect its generalization ability in unseen scenarios. Thirdly, as continual learning tasks accumulate, the parameter and storage overhead of the MoE structure may gradually increase, posing a potential burden for practical deployment.

---

> ### Author Rebuttal · Authors · 2026-03-31
>
> Thanks so much for the positive feedback and for pointing out those critical details on compute and generalization. You raised some great points. Here is our breakdown:
>
> **1. Extra computation and storage overhead of routing / PTL.**
>
> Our goal wasn't just to throw more parameters at the problem to buy better performance. We really wanted to keep the overhead for task-aware expansion and replay-free routing as low as possible. Here’s how the math looks for compute:
> * A single VAE inference takes about 100 MFLOPs. Even with 8 iterations, that’s only about 0.8 GFLOPs.
> * For a typical input (shape `[1, 768, 2048]`), adding one LoRA expert costs about 25 MFLOPs, and one router takes about 12 MFLOPs.
> * Even if we maxed things out and added 1 expert and 1 router to all 32 layers, the extra compute is just (25 + 12) × 32 = 1.184 GFLOPs.
> * To put that in perspective, LLaVA-1.5-7B's vision encoder uses about 183.16 GFLOPs per 336×336 image. Our extra compute is only about 0.4% of that.
>
> As for storage, PTL doesn't need us to save a whole separate backbone. We just piggyback on LLaVA's native features, so there’s zero extra cost for a feature extractor. On average, we only added 43.96M trainable parameters per task, insignificant compared to the 7B backbone.
>
> **2. Details on the Top-K setting.**
>
> We totally agree we didn't explain this well enough in the text. In all our experiments, `top-k` isn't a static number; it's dynamically set to $k=\lfloor N/4 \rfloor$, where $N$ is the total number of experts. It naturally scales up as the model grows.
>
> To clear up any doubts, we ran a quick test across the 8 tasks to compare our dynamic $k$ against fixed values like $k=2$ or $k=3$. As the table below shows, the dynamic approach does a much better job balancing performance with the inference cost of calling experts:
>
> | Dataset | scienceqa | textvqa | imagenet | gqa | vizwiz | ref | vqav2 | ocrvqa |
> | :--- | :--- | :--- | :--- | :--- | :--- | :--- | :--- | :--- |
> | **topk=2 (imm)** | 78.82 | 59.81 | 96.53 | 56.30 | 57.23 | 25.14 | 65.28 | 61.74 |
> | **topk=3 (imm)** | 79.89 | 59.98 | 97.07 | 56.26 | 58.16 | 25.77 | 65.58 | 62.92 |
> | **ours (imm)** | 79.01 | 59.94 | 96.85 | 56.43 | 57.44 | 25.63 | 65.15 | 62.01 |
>
> *(Note: "imm" is immediate performance).*
>
> We’ll make sure the formula is clearly stated in the revision and add a master table for all these hyperparameters (like $\alpha$, $T$, and $N_s$) so they're easy to find.
>
> **3. Further clarification on zero-shot generalization.**
>
> We wanted to share two quick findings that show CoPE actually works outside of just the seen-task setting:
> * For zero-shot, we took the checkpoint right after training the first 6 tasks and tested it straight on the 7th (VQAv2) without any training. We hit 48.32% accuracy (for context, the Immediate/Final results were 65.15% / 56.40%).
> * When we looked at what PTL was actually doing during that test, we saw that about 80% of the VQAv2 samples were routed to TextVQA's router and experts. This tells us that PTL doesn't just guess randomly on unseen tasks, it actively hunts for the closest semantic match, which helps preserve some zero-shot ability.
>
> **4. Parameter growth as tasks accumulate.**
>
> Parameter bloat is a real problem for continual MoE methods. But that's exactly the reason we purposed CoPE. Instead of blindly expanding every layer, we only add experts where the probe tells us we absolutely need them, keeping parameter growth at a low level. Compared to the MoExtend codebase, we got better or similar performance with fewer experts on most tasks. So while we can't magically stop parameter growth entirely, CoPE keeps it on a very tight leash without sacrificing our anti-forgetting setup.

---

> > ### Author Rebuttal · Reviewer_hv34 · 2026-04-01
> >
> > Thanks for the detailed clarifications.
> >
> > My main concerns were the extra computation introduced by MoE and the choice of $k$. The authors show that the overhead is only 0.4% of visual encoder, which is acceptable. The dynamic setting $ k = \lfloor N/4 \rfloor $ is also well justified and alleviates concerns about reduced activation with more experts. The added zero-shot results and PTL routing analysis further support generalization.
> >
> > Overall, my concerns are addressed, and I am willing to raise my score to 5.

---

> > > ### Author Response · Authors · 2026-04-04
> > >
> > > Dear Reviewer hv34,
> > >
> > > Thank you for the positive feedback and for raising the score. We are glad that our rebuttal clarified your concerns regarding computational overhead, the dynamic $k$ setting, and zero-shot generalization. We also appreciate your recognition of our FLOPs breakdown and PTL routing analysis.
> > >
> > > We will ensure all discussed details—including the adaptive $k$ logic and the hyperparameter table—are fully integrated into the revised manuscript to improve its clarity.
> > >
> > > Best regards,
> > >
> > > Authors

---

### Decision · Program_Chairs · 2026-04-30

**Decision:**

Accept (regular)

**Comment:**

I recommend acceptance.

Reviewers found the paper technically sound and supported by solid empirical evidence. The paper’s main contribution is a coherent probe-guided expansion framework for replay-free continual adaptation of large VLMs with convincing gains in the main evaluated setting. Reviewer hv34 found that the rebuttal addressed concerns about MoE overhead, expert selection, and PTL behavior, while Reviewer khSU viewed the work as a well-integrated adaptive MoE system with meaningful empirical gains, though with moderate rather than deep methodological novelty. Reviewers WQ13 and qP2K also indicated that their concerns were addressed. The authors are encouraged to integrate the key rebuttal clarifications into the final version.